# Universal Lifespan Trajectories of Source-Space Information Flow Extracted from Resting-State MEG Data

**DOI:** 10.3390/brainsci12101404

**Published:** 2022-10-18

**Authors:** Stavros I. Dimitriadis

**Affiliations:** 1Neuroscience and Mental Health Research Institute (NMHI), College of Biomedical and Life Sciences, Cardiff University, Maindy Road, Cardiff CF24 4HQ, Wales, UK; dimitriadiss@cardiff.ac.uk; 2Cardiff University Brain Research Imaging Centre (CUBRIC), School of Psychology, College of Biomedical and Life Sciences, Cardiff University, Maindy Road, Cardiff CF24 HQ, Wales, UK; 3MRC Centre for Neuropsychiatric Genetics and Genomics, Division of Psychological Medicine and Clinical Neurosciences, Cardiff School of Medicine, Cardiff University, Maindy Road, Cardiff CF24 4HQ, Wales, UK; 4Neuroinformatics Group, School of Psychology, College of Biomedical and Life Sciences, Cardiff University, Maindy Road, Cardiff CF24 4HQ, Wales, UK; 5MRC Integrative Epidemiology Unit (IEU), University of Bristol, Queens Road, Bristol BS8 1QU, Wales, UK; 6Department of Clinical Psychology and Psychobiology, Faculty of Psychology, University of Barcelona, Passeig de la Vall d’Hebron, 171, 08035 Barcelona, Spain; 7Institut de Neurociències, University of Barcelona, Campus Mundet, Edifici de Ponent, Passeig de la Vall d’Hebron, 171, 08035 Barcelona, Spain; 8Integrative Neuroimaging Lab, 55133 Thessaloniki, Macedonia, Greece

**Keywords:** magnetoencephalography, resting state, information flow, symbolic transfer entropy, atlas-based source localization, development, intrinsic coupling modes, universal brain age index

## Abstract

Source activity was extracted from resting-state magnetoencephalography data of 103 subjects aged 18–60 years. The directionality of information flow was computed from the regional time courses using delay symbolic transfer entropy and phase entropy. The analysis yielded a dynamic source connectivity profile, disentangling the direction, strength, and time delay of the underlying causal interactions, producing independent time delays for cross-frequency amplitude-to-amplitude and phase-to-phase coupling. The computation of the dominant intrinsic coupling mode (DoCM) allowed me to estimate the probability distribution of the DoCM independently of phase and amplitude. The results support earlier observations of a posterior-to-anterior information flow for phase dynamics in {α_1_, α_2_, β, γ} and an opposite flow (anterior to posterior) in θ. Amplitude dynamics reveal posterior-to-anterior information flow in {α_1_, α_2_, γ}, a sensory-motor β-oriented pattern, and an anterior-to-posterior pattern in {δ, θ}. The DoCM between intra- and cross-frequency couplings (CFC) are reported here for the first time and independently for amplitude and phase; in both domains {δ, θ, α_1_}, frequencies are the main contributors to DoCM. Finally, a novel brain age index (BAI) is introduced, defined as the ratio of the probability distribution of inter- over intra-frequency couplings. This ratio shows a universal age trajectory: a rapid rise from the end of adolescence, reaching a peak in adulthood, and declining slowly thereafter. The universal pattern is seen in the BAI of each frequency studied and for both amplitude and phase domains. No such universal age dependence was previously reported.

## 1. Introduction

The electrophysiological activity of the brain is dominated by rhythmic activity over a wide range of frequencies from below 1 Hz to δ (1–4 Hz) [1,2], θ (5–8 Hz), α (9–12 Hz), σ (12–16 Hz), β (16–35 Hz), γ (35–100 Hz), and even higher frequencies. These frequencies recur across levels—from the neural membrane [3] to a macroscale of EEG/MEG—and contribute to cognitive processes [4,5,6]. Lopes da Silva et al. (1980) suggest that frequency is used locally and globally for complex multiplexing, organization, and coordination of brain activity in time. This is confirmed by analysis using both power spectra [1,2,7,8] and connectivity measures during different cognitive tasks [7,9,10,11,12,13]. 

The exchange of information between brain areas, some close and others distant, poses a complicated multiplexing problem that the brain solves using cross-frequency in-teractions in multiple simultaneous brain rhythms [14]. Evidence for these has been iden-tified in different species over a range of brain sizes and anatomical/functional connectiv-ity complexities [15,16,17]. In summary, temporal parcellation of function is mediated by communication between spatially distributed nodes, with oscillations guiding dynamic integration of local processing and flexibly adjusting to the changing cognitive demands of successive tasks [18,19,20]. 

Brain areas can communicate at zero phases if their output is synchronized to an os-cillation cycle, whereas inputs arrive within the excitatory phase of the same cycle. A cycle duration directly linked to a given oscillation frequency can support communication with a fixed maximum time delay [21], with delays increasing (frequency decreasing) as the distance between brain areas increases [22,23]. 

The human brain is a complex system [1,2,3] consisting of interconnected functional units at the macroscopic scale [4] with specific information processing capabilities [5]. Cognitive functions can be supported by the coordinated activity of these spatially distinct units, whereby the oscillatory nature of these interactions can provide the mechanism [6,7,8,9]. 

The brain is an extremely complex system [24,25,26] containing, at the macroscopic scale, interconnected functional units [27] with more or less specific information-processing capabilities [28]. However, cognitive functions require the coordinated activity of these spatially separated units, whereby the oscillatory nature of the neuronal activity may provide a possible mechanism [16,29,30,31]. The activity of these functional units oscillating on a preferred frequency is coordinated via cross-frequency interactions [5]. The exploration of these interactions, in terms of frequency content, strength, directionality, and the time delay is more than necessary for a better understanding of how the brain functions and dysfunctions under both normal and abnormal conditions, respectively. 

Functional interactions may be investigated by statistical dependencies between time series of brain activity at different regions with a frequency content [10]. These interac-tions are so-called functional connectivity and effective connectivity for causal dependen-cies. Such interactions have been explored across large-scale networks in magnetoenceph-alography (MEG) and electroencephalography (EEG) [9,32,33]. Until now, only one study has explored the directionality of interactions across large-scale networks in the phase domain following a within-frequency analysis [33]. The authors of this study adopted a data-driven estimator, so-called phase transfer entropy (PTE), to explore the directionality of frequency-dependent, large-scale, MEG source-reconstructed, resting-state activity. At higher frequencies (8–30 Hz), they showed dominant posterior-to-anterior patterns of in-formation flow in the parieto-occipital lobe toward frontal areas. In contrast, a pattern of anterior-to-posterior flow was found in the θ band, whereas the senders of information in the α_2_ band were also often receivers of information in the θ band, suggesting a fre-quency-specific loop of information flow in the human brain. Causal dependencies should be also explored in the amplitude domain. In our previous study, we designed delay sym-bolic transfer entropy (dSTE) to explore directionality, strength, and time delay between two time series from EEG sensor locations functioning at different frequencies [34]. We revealed effective interactions between frontal^θ^ (F^θ^) and PO^α2^ consistently across the diffi-cult levels of a mental arithmetic task. 

The majority of functional neuroimaging studies have explored the effective connec-tivity patterns between whole-brain parcellated brain areas, focusing on within-frequency coupling interactions [35]. However, there are numerous studies supporting the existence of true cross-frequency coupling (CFC) interactions between brain areas that coexist with between-frequency coupling (BFC) interactions [36,37]. The authors’ studies integrating both types of interactions assumed that these interactions coexist in every temporal seg-ment and across every pair of brain areas. They tabulated all these coupling strengths in a multilayer network of (size number of frequencies x number of brain areas)[36,37]. The construction of a multilayer network where both BFC and CFC coexist among every stud-ied frequency and set of brain areas is an overestimation of what really happens in the human brain in every condition. For that reason, we designed a statistical framework that detected the so-called dominant coupling mode between every pair of brain areas in a specific temporal segment. This dominant coupling mode could be either BFC within a specific frequency band, CFC between a specific frequency pair, or with no interaction [5].

The dominant coupling modes model (DoCM) provides a unifying framework for capturing the dynamics of intrinsically generated neuronal interactions at multiple spatial and temporal scales [5]. DoCM can be captured in both amplitude and phase domains, as well as across both within-frequency and cross-frequency interactions. All these potential coupling modes between two time series at distinct anatomical sites that coexist in the resting state are so-called intrinsic coupling modes (ICMs). Using MEG and EEG, it is pos-sible to study intrinsic coupling modes (ICMs) across a broad range of time scales and in a spectrally resolved manner [38,39]. ΙCMs are important features of ongoing brain activ-ity that show rich spatiotemporal distribution and contain information that influences cognition [5]. Here, we will focus on two basic CFC types: one that arises from phase cou-pling of band-limited oscillations and a second that arises from the amplitude coupling of fluctuations of band-limited oscillations [5,40]. Many studies have demonstrated that studying ICMs with electrophysiology can contribute complementary information to fMRI with superior temporal and spectral resolution [41,42]. We showed how DoCM can be performed in functional neuroimaging modalities in our previous studies [7,12]. 

It is now accepted that cross-frequency coupling in ongoing activity [43,44,45] contains information that cannot be captured by fMRI. Furthermore, patterns of resting-state cross-frequency coupling extracted from MEG are significantly different compared to controls in dyslexia [46], mild traumatic brain injury [47], and mild cognitive impairment [48]. 

In the present study, apart from repeating the same analysis as in [33], we aimed to explore the directionality, strength, and time delay between large-scale, resting-state, source-reconstructed networks in both in-phase (PTE) and amplitude domains (dSTE). In addition, we adopted both estimators to explore the aforementioned features of both within-frequency and cross-frequency interactions for the first time in the literature. This procedure can reveal the dominant coupling modes per pair of brain regions and across epochs with our dominant coupling modes model (DoCM). Our analysis is applied to a lifespan open MEG cohort with the main goal of identifying possible age-dependent trends. We analyzed a large number of epochs compared to only one large epoch in [33] in both within- and between-frequency coupling with two estimators and assessed the repeatability of the extracted features in a repeat cohort. 

In this work, we identify consistent information flow patterns across ages using delay symbolic transfer entropy (dSTE) [34,49] and phase transfer entropy (PTE) [50]. Four dis-tinct developmental changes are documented: (1) age-dependent trends in the infor-mation flow between anterior and posterior parts of the brain at some of the key brain rhythms that are similar for amplitude and phase dynamics; (2) meantime lags (time de-lays) between regional brain activity within and between brain rhythms that are similar for amplitude and phase dynamics through the ages; (3) characterization of the prominent coupling between every pair of source time series under the notion of DoCM; and (4) a universal developmental history represented by a ratio of changes of DoCM. 

## 2. Materials and Methods

### 2.1. Subjects

In this study, we used the main analysis of resting-state MEG data from the Open Access Omega Project [51]. We selected 103 healthy control subjects based on small head movements (less than 4 mm) and gender balance (51 females and 52 males), as well as uniform spread in the age range of 18–65 years (see Table 1). 

We independently analyzed a separate data set of ten healthy young adults (five women aged 24.4 ± 1.5 years, and five men aged 25.3 ± 1.7 years) recorded with the MEG CTF 275 sensor system at the CUBRIC Neuroimaging Centre of Cardiff University. In this experiment, MEG data were obtained twice from each subject, using, in each case, an eyes-closed resting-state task lasting 5 min. The two recording sessions were held a week apart from each other. The experiment was performed under ethical approval from the School of Psychology. 

### 2.2. MEG-MRI Recordings

In this study, we analyzed MEG and MRI data sets from the OMEGA (Open MEG Archive) repository. Resting-state, eyes-open activity was recorded with a minimum du-ration of five minutes. MEG data were collected at the BIC and the Université de Montréal on identical CTF whole-head MEG systems (VSM MedTech Inc., Coquitlam, BC, Canada) consisting of 275 first-order, axial-gradiometer coils and third-order gradient correction to subtract background interferences with passive magnetic shielding. Fiducial and head-shape information obtained through 3D digitization during subject preparation, as well as head-motion information collected via head-positioning coils, is available for all participants [51]. We excluded any subject with more than 4 mm head movement. 

The data were first whitened and reduced in dimensionality using principal compo-nent analysis with a threshold set to 95% of the total variance [52]. The statistical values of kurtosis, Rényi entropy, and skewness of each independent component were used to eliminate ocular and cardiac artifacts. Specifically, a component was deemed artefactual if more than 20% of its values after normalization to zero mean and unit variance were outside the range of (2, +2) [47,53,54]. The artifact-free, multichannel MEG, resting-state recordings were then entered into the beamforming analysis (see Section 2.3).

### 2.3. Beamforming

An atlas-based beamformer approach was adopted to project MEG data from the sensor level to the source space independently for each brain rhythm. The following brain rhythms were studied: δ (1–4 Hz), θ (4–8 Hz), α1 (8–10 Hz), α2 (10–13 Hz), β (13–30 Hz), and γ (30–45 Hz). First, the coregistered MRI was spatially normalized to a template MRI using SPM8 [55]. The automated anatomical labeling (AAL) atlas was used to anatomi-cally label the cortical voxels in a subject’s normalized, coregistered MRI [56]. After in-verse transformation to the patient’s coregistered MRI [57], neuronal activity in the atlas-labeled cortical voxels was reconstructed using the LCMV source localization algorithm as implemented in Fieldtrip and transformed to the MNI template [58]. 

The beamformer sequentially maps the activity for each voxel in a predefined grid covering the entire cortex (spacing 6 mm) by weighting the contribution of each MEG sensor to a voxel’s time series, a procedure routinely used to project the sensor activity to the cortical activity. Each region of interest (ROI) in the atlas contains many voxels, and the number of voxels per ROI differs. To obtain a single representative time series for every ROI, we defined a functional centroid ROI representative of ROI as the functional interpolated activity from the voxel time series within each ROI. Specifically, we estimated a functional connectivity map between every pair of source time series within an ROI (Equation (1)) using correlation (Equation (2)); then, we estimated the strength of each voxel from the connectivity map within the ROI (Equation (3)). Finally, we normalized each strength by the sum of strengths (Equation (4)). The procedure produces a weight for each voxel within each ROI satisfying the condition that their sum is unity. Finally, summing the sum of the weighted time series over the voxels affords the representative time series of each ROI (Equation (5)). This procedure is similar in spirit to the interpola-tion of a bad channel in an EEG/MEG grid by the activity of the neighboring sensors. The whole procedure was applied independently to every quasi-stable temporal segment de-rived from the settings of the temporal window and stepping criterion. 

Equations (1)–(5) document the steps for this functional interpolation.
(1)$$ROI^{map}\in R^{voxels\;x\;samples},\;voxels\;\in no\;of\;voxel\;timeseries\;within\;each\;ROI$$
(2)$$S^{Voxels}=\overset{Voxels}{\underset{k=1}{\sum }}\overset{Voxels}{\underset{t=k+1}{\sum }}\left|corr\left(ROI_{k}^{map}\left(t\right),ROI_{l}^{map}\left(t\right)\right)\right|,\;\;S^{Voxels}\;\in ROI\;X\;ROI$$
(3)$$SS_{k}=\overset{Voxels}{\underset{k=1}{\sum }}corr\left(k,:\right),\;SS\;\in 1\;x\;ROI$$
(4)$$W_{k}=\frac{SS_{k}}{\sum_{k=1}^{Voxels}SS_{k}}$$
(5)$$ROI^{activity}=\overset{Voxels}{\underset{k=1}{\sum }}ROI_{k}^{time\;series}*W_{k}$$

Figure 1 provides a graphical representation of the preprocessing steps in Equations (1)–(5).

For dynamic source connectivity analysis, we used a four-second-long window, yielding 15 (epochs per min) × 5 (min) = 75 non-overlapping epochs [33]. The same epoching approach was used for every frequency band to explore direction and time delays within and between frequency bands and in both amplitude/phase domains. The procedure described above for extracting a single representative time series for every ROI was adopted independently per epoch and frequency bands for every subject.

The strength, direction, and time lag of the direction of information flow were estimated between the 78 cortical regions in the automated anatomical labeling (AAL) [33] atlas using directed phase entropy (dPTE) and directed symbolic transfer entropy (dSTE).

### 2.4. Overview of the Methodology

In the present study, we adapted two estimators, namely PTE and dSTE, with the main aim of revealing causal interactions between phase and amplitude frequency-dependent interactions from every possible pair of virtual sensors. The main goals of this research study were the following: (a) We adopted a similar analysis using PTE as in a previous study [45] to reveal possible lifespan trends in terms of causal interactions between frequency-dependent time series in the phase domain. (b) We adopted the same analysis using a proper estimator tailored to reveal causal interactions between frequency-dependent time series in the amplitude domain (dSTE). (c) We aimed to reveal dominant coupling modes and their probability distributions (PD) by exploring causal relationships between virtual sensors oscillating in the same (within frequencies) and different frequencies (cross frequencies). (d) We defined a brain age index based on the ratio of PD of cross-frequency couplings and within-frequency couplings. (e) We untangled developmental trends of time delay both in within-frequency couplings and between dominant coupling modes.

In the present study, apart from re-evaluating the findings from a previous study [45] using a lifespan cohort and a large enough number of epochs, we attempted, for the first time in the literature, to explore causal interactions in both amplitude and phase domains both within and between frequencies. Previous studies, for simplicity, independently ex-plored causal interactions per frequency band, ignoring cross-frequency interactions. However, brain regions communicate and exchange information with each other with a preferred coupling mode, which can manifest as either within-frequency coupling or be-tween-frequency coupling. For that reason, we developed a dominant coupling modes model (DoCM), which serves as a way to untangle the dominant coupling mode between every pair of brain regions [7]. Adopting these two estimators, we can also reveal the fre-quency-dependent time delays both for within- and between-frequency coupling modes. Time delay is an important feature of spatiotemporal causal interactions that shape the information flow across anatomical space and time, overcoming any neurophysiological and anatomical constraints. 

Figure 2 exemplifies how our DoCM works. In the present study, following our DoCM, we computed the causal interactions between every possible pair of virtual sensors using dPTE (Figure 2A) and dSTE (Figure 2B). Both estimators were employed to quantify the strength and time delay of causal interactions, both within frequencies and between frequencies (cross-frequency). Then, by adopting a surrogate analysis, we revealed statistically significant interactions that deviate from chance. Surrogate analysis revealed the preferred interaction that is accompanied by the strength of the interaction, the dominant coupling mode, and the time delay. In the example in Figure 2, dPTE and dSTE estimators reveal that the first ROI drives the second ROI in both phase and amplitude domains, whereas the time delay is 10 ms and 86 ms, respectively. For further details, see Section 2.6 and Figure 3. 

### 2.5. Information Flow with dSTE and dPE

#### 2.5.1. Delay Symbolic Transfer Entropy (dSTE), Delay Phase Transfer Entropy (dPTE), and Significance Test

##### Delay Symbolic Transfer Entropy (dSTE)

In principle, asymmetric dependences between coupled systems can be detected with measures that share some of the properties of mutual information [59] and take into ac-count the dynamics of information transport. Transfer entropy [60], which is related to the concept of Granger causality [61], has been proposed to effectively distinguish driving and responding elements and to detect asymmetry in the interaction between subsystems. By appropriate conditioning of transition probabilities, this quantity is superior to the standard time-delayed mutual information, which fails to distinguish information that is actually exchanged from shared information due to common history and input signals. Various techniques have been proposed to estimate transfer entropy from observed data. However, most techniques make considerable demands on the data, require fine-tuning of parameters, and are highly sensitive to noise contributions, which limits the use of transfer entropy to field applications [62,63]. 

Symbolic transfer entropy (STE) was proposed to overcome the limitations of opti-mized parameters needed for the estimation of transfer entropy [64]. In the present study, we adopted the Neural Gas algorithm [65] as an appropriate technique to create a com-mon codebook for a multichannel data set [66]. 

In the present study, dSTE was applied in the amplitude domain. Information flow was estimated independently between every pair of ROIs oscillating at the same fre-quency (BFC interactions) or at different frequencies (CFC interactions) and across all tem-poral segments. Below, we describe the algorithmic steps of dSTE estimation that were adopted in the amplitude domain. 

Here, we describe the algorithmic steps with which we transcribed the temporal dy-namics from any pair of virtual sensors into two distinct symbolic time series that share a common codebook (set of symbols). The size and content of the codebook are data-de-pendent and estimated every time causal relationships are inferred from a pair of recorded signals. The associated computational burden is kept low thanks to the unsupervised al-gorithm employed (i.e., Neural Gas) [34]. 

Given the signals ^A^x_t_ and ^B^x_t_ from a pair of channels A and B, time-delay vectors are first reconstructed from each time series. These vectors take the form of $x_{t}=\left\{x_{t},x_{\left(t+\tau \right)},\ldots ,x_{\left(t+\left(m-1\right)\tau \right)}\right\}$, where the embedding dimension (*τ*) denotes the time lag, and t = 1, 2, …, T runs over the time points.

Then, the two individual sequences of time-delay vectors are collectively gathered in data matrices:^A^X_[Txm]_ = [^A^X_1_ | ^A^X_2_| … | ^A^X_T_] & ^B^X_[Txm]_ = [^B^X_1_ | ^B^X_2_| … | ^B^X_T_] (6)

Next, the two trajectories are brought to a common reconstructed state space by forming the overall data matrix:^AB^X_[2Txm]_ = [^A^X | ^B^X](7)

The partition of all the tabulated m-dimensional vectors into groups of homogenous patterns is the most direct way to summarize the temporal variations in the activations of these two subsystems and describe them with a common vocabulary.

In our approach, a codebook of *k* code vectors is designed by applying the NG algorithm to the data matrix (^AB^**X**), which is of size [~2T × m]. The NG algorithm is an artificial neural network model that converges efficiently to a small number (k << T) of codebook vectors ({M_i_}_i=1:k_) using a stochastic gradient descent procedure with a soft-max adaptation rule that minimizes the average distortion error [65].

In the encoding stage, each of the 2*T* vectors is assigned to the nearest code vector. By replacing the original vectors with the assigned code vectors, we can rebuild the two vectorial time series with a measurable error. If we denote the reconstructed (i.e., decoded) version of the vectorial time series as ^AB^X^rec^ (t), we can estimate the fidelity of the overall encoding procedure with the following index, which is the total distortion error divided by the total dispersion of the original vectors:(8)$$n^{Distortion}=\frac{\sum_{t=1}^{2T}\|X^{AB}\left(t\right)-X_{rec}^{AB}\left(t\right){\|}^{2}\;}{\sum_{t=1}^{2T}\|X^{AB}\left(t\right)-{\bar{X}}^{2}\|\;},\;\bar{X}=\frac{1}{2T}\;\sum_{t=1}^{2T}X^{AB}\left(t\right)$$

The smaller the $\;n^{Distortion}$, the better the encoding. This index gets smaller with an increase in *k*, reaching a plateau for a relatively low value of *k*. In the present study, we considered encoding to be acceptable if it was produced with the lowest *k* value that satisfied the condition that $n^{Distortion}$ should be less than 5%. Hence, we repeatedly quality. In this way, we defined the optimal *k_o_*, which in turn defined the codebook for use in the subsequent symbolization scheme. At the vector-quantization stage, each vector of **^A^X** and **^B^X** is assigned (according to the nearest-prototype rule) to the most similar among the derived code vectors ({M_i_}_i=1:ko_). This step completes the mapping of the original time series to two symbolic time series (^A^s_t_ and ^B^s_t_), t=1, 2,…,T, which, in mathematical notation, reads as follows:(9)$$\begin{array}{c} \left[X_{t}^{A},X_{t}^{B}\;\right]\in R^{2}\;\\ X_{t}^{A}\;\overset{VQ}{\to }\;M_{j_{1}}\;\in \;{\left\{M_{i}\right\}}_{i=1}^{k_{o}},M_{i}\in R^{m}\;\;\;\;,\;\;\;\;\;X_{t}^{B}\;\overset{VQ}{\to }\;M_{j_{2}}\;\in \;{\left\{M_{i}\right\}}_{i=1}^{k_{o}},M_{i}\in R^{m}\\ X_{t}^{A}\to S_{t}^{A}=j_{1}\left(t\right),X_{t}^{B}\to S_{t}^{B}=j_{2}\left(t\right),j_{1},j_{2}\;\in \left\{1,2,\ldots ,k_{o}\right\} \end{array}$$

In the derived symbolic time series, the temporal dynamics of a pair of neural subsystems are encoded as transitions among adaptively defined (i.e., data-dependent) symbols.

We adopted the Ragwitz criterion to optimize the embedding dimension (*d*) and the embedding delay (*τ*) [67]. Optimality of embedding refers to a minimal prediction error for future samples of the time series. The Ragwitz criterion predicts the future of a signal based on estimates of the probability densities of future values of its nearest neighbors after embedding. The adopted method is based on the minimization of mean squared prediction error [67,68].

The *m* parameter ranged from 7 to 10, and the τ parameter ranged from 3 to 9 for the entire set of subjects.

The objective criterion of the best fitting of the algorithm was the distortion error, which was set as in the amplitude domain ($n^{Distortion}$ should be less than 5%).

##### Quantifying Effective Connectivity with dSTE

Providing a pair of symbolic sequences (^A^s_t_ and ^B^s_t_), the relative frequency of symbols can be used to estimate joint and conditional probabilities and to define STE as follows:(10)$$STE_{BA}=\sum p\left(S_{t+\delta }^{A},S_{t}^{A},S_{t}^{B}\right)log\frac{p\left(S_{t+\delta }^{A}/S_{t}^{A},S_{t}^{B}\right)}{p\left(S_{t+\delta }^{A}/S_{t}^{A}\right)}\;\;\;$$
where the sum runs over all symbols, and *δ* denotes a time step.

Effective connectivity is defined as ‘the influence one system exerts over another [61,69]. In the context of brain networks, effective connections are directed from one brain area to another. To account for the time delay between brain activation signals from distant areas, we modified the previous definition:(11)$$dSTE_{BA}=STE_{BA}\left(d\right)=\sum p\left(S_{t+1}^{A},S_{t}^{A},S_{t+1-d}^{B}\right)log\frac{p\left(S_{t+1}^{A}/S_{t}^{A},S_{t+1-d}^{B}\right)}{p\left(S_{t+1}^{A}/S_{t}^{A}\right)}\;$$
where *d* is the time delay between the driving and the driven systems. The log is with base 2; thus, STE_BA_ is given in bits. STE_AB_ is defined in complete analogy. The directionality index (DdSTE_AB_ = dSTE_AB_ − dSTE_BA_) quantifies the preferred direction of information flow and achieves positive values for unidirectional couplings with A as the driver and negative values for B driving A. For symmetric bidirectional couplings, ΔdSTE is approximately zero. The formulation of TE with a time delay was first proven to be correct in a recent study, which presented a robust method for neuronal interaction delays [70].

To detect significant causal interactions between two brain regions (considered subsystems A and B), we adopted a well-known technique described by Chavez et al., 2003 [63], Verdes, 2005 [62], Lizier et al., 2011 [71], and Vicente et al., 2011 [72]. The original approach was developed for TΕ but can easily be applied to its symbolic counterpart. The null hypothesis (H_0_) of the test is that the state transitions ($X_{n}^{A}\;\to \;X_{n+1}^{A}$) of the destination system (A) have no temporal dependence on the states of the source system (B). We form a distribution of dSTE measurements ${\left\{dSTE_{BA}^{H_{o}}\;\right\}}_{r=1:1000}\;$ under this condition by repeatedly applying the following algorithmic steps:

Step_i: Generate a surrogate time series by permuting the elements of the source symbolic time series, ^B^s_t_;

Step_ii: Estimate an instantiation of the ‘randomized’ dSTE_BA_ using ^A^s_t_ and the surrogate ^B^s_t_ in Equation (11).

We can then determine a one-sided *p*-value that corresponds to the likelihood that the actual observed value, namely observed dSTE_BA_, is within the range of values of the distribution (${\left\{dSTE_{BA}^{H_{o}}\;\right\}}_{r=1:1000}$). This can be achieved by directly estimating the proportion of ‘randomized’ dSTE_BA_ that are higher than the observed dSTE_BA_ value [66,71]. The false-discovery rate (FDR) method [73] was employed to control for multiple comparisons (across all possible pairs of ROIs), with the expected proportion of false positives set to *q* ≤ 0.01. Finally, the dSTE mode that characterized a specific pair of ROIs was determined based on the highest statistically significant (dSTE) value from surrogates. FDR correction was applied at a brain network level (ROIs × ROIs) independently for each epoch, as well as within-frequency and cross-frequency pairs and subjects.

##### Quantifying Effective Connectivity with Delay Phase Transfer Entropy (dPTE)

Transfer entropy can be estimated from the time series of the instantaneous phases (PTE) at a low computational cost [50]. In the case of the phase domain, phase dynamics were extracted from the frequency-dependent, ROI-based time series via the Hilbert transform. Similarly, as in the amplitude domain, information flow in the phase domain was estimated independently based on the Hilbert-transformed, ROI-based time series derived from brain activity oscillating at the same frequency (BFC interactions) or at different frequencies (CFC interactions) and across all temporal segments.

If the uncertainty of a target signal *Y* at a delay *δ* is expressed in terms of Shannon Entropy, then the Transfer Entropy (TE) from source signal *X* to target signal *Y* can be expressed as:
(12)$$TE_{xy}=\sum p\left(Y_{t+\delta },Y_{t},{Χ}_{t}\right)log\frac{p\left(Y_{t+\delta }/Y_{t},{Χ}_{t}\right)}{p\left(Y_{t+\delta }/Y_{t}\right)}$$
𝑤here the definition of Shannon entropy is given by 𝐻(𝑌𝑡 + 𝛿) = −Σ𝑝(𝑌𝑡 + 𝛿)𝑙𝑜𝑔𝑝(𝑌𝑡 + 𝛿), was used, and the sum runs over all discrete time steps *t*. The estimation of probability is time consuming and for that reason, Staniek and Lehnertz proposed the estimation of transfer entropy over converted time series into symbols [64]. Under the same framework, time series can be described in terms of instantaneous phases as of their amplitudes [74]. Transfer entropy can be estimated from the instantaneous phases of two time series at a low computational cost [50]. 

Dropping the subscript t for clarity and using the fact that *p*(*Y_δ_*,*Y*) = *p*(*Y**_δ_*) *p*(*Y*), the PTE becomes:(13)$$PTE_{xy\;}=\sum p\left(Y_{\delta }\right)p\left(Y\right)p\left(X\right)log\left(\frac{p\left(Y,X\right)}{p\left(\Upsilon \right)}\right)\;$$
where the probabilities are obtained by building histograms of occurrences of single, paired, or triplet phase estimates in an epoch [50].

The number of bins in the histograms was set as $e^{0.626+0.4ln\left(N_{s}-\delta -1\right)}$ [74]. Finally, δPTE was normalized according to the following formula:(14)$$dPTE_{xy}=\frac{PTE_{xy}}{PTE_{xy}+PTE_{yx}}\;$$

The value of *dPTE_xy_* ranges between 0 and 1. When information flows preferentially from time series X to time series Y, 0.5 < *dPTE_xy_* ≤ 1. When information flows preferentially toward X from Y, 0 ≤ *dPTE_xy_* < 0.5. In the case of no preferential direction of information flow, *dPTE_xy_* = 0.5.

For dPTE, we randomly shuffled the time index of the epochs of 4 s between every pair of ROIs to create a surrogate-based dPTE distribution. For example, we estimated dPTE between ROIs 1 and 2 by employing the 1st epoch of ROI 1 with the 2nd epoch of ROI 2. Out of 75 × 75 – 75 = 5550 possible combinations of epochs, we employed 1,000 combinations leading to 1000 surrogates, and following the same statistical analysis as with dSTE. 

#### 2.5.2. Common Normalization of dSTE and dPE

In the present study, we adopted dSTE as the proper estimator to explore information flow between the activity of brain areas in the amplitude domain both for BFC and CFC interactions [34,49,75]. Because dSTE does not have an upper boundary like the well-known connectivity estimators, we defined the normalized version of dSTE as follows:(15)$$\Delta dSTE_{ij}=\frac{dSTE_{ij}-dSTE_{ji}\;}{dSTE_{ij}+dSTE_{ji}}\;$$

For the estimation of information flow based on phase dynamics, the Hilbert transform of the frequency-dependent, representative time series per ROI was used both for BFC and CFC interactions. Then, we adopted phase entropy (PTE) [50] using the settings described by Hillebrand et al., 2016 [33] but applying the same normalization as above:(16)$$\Delta dPTE_{ij}=\frac{dPTE_{ij}-dPTE_{ji}\;}{dPTE_{ij}+dPTE_{ji}}$$

For both estimators, the value of *Δ**dSTE_ij_*/*Δ**dPTE_ij_* ranges between −1 and 1. The range of values is interpreted as follows:

+1 if information flows exclusively from i →j;

–1 if information flows exclusively from j →i; and

0 if information flows equally well between *i* and *j*,

where *i* and *j* refer to brain areas, such as anterior and posterior.

Both definitions are measures of the proportion of information flow in each direction in the two ROIs and not the quantity of information flow.

### 2.6. Time-Lag Estimation

The representative time lag per pair of MEG source epochs and across the cohort was estimated via surrogate analysis, and the appropriate statistical analysis was followed for both estimators (see ‘Section 2.5’ and [34,49,50,75]).

The dSTE/dPTE were estimated by shifting one of the time series concerning the other at lags corresponding to ± 0.5 epoch lengths (where epoch length denotes the length of an epoch in seconds). We then identified, for each pair of time series, the maximum Δ*dSTE*/Δ*dPTE* value among those derived from the set of examined lags. Employing the precomputed dSTE/dPTE values over each of the examined lags (dSTE^lags^/dPTE^lags^), we estimated the z-score of maximum Δ*dSTE*/Δ*dPTE* based on the mean and standard deviation of dSTE^lags^/dPTE^lags^. Finally, for each pair of MEG ROIs time series and epoch in the entire set of cohorts, we assigned a time-lag estimation for Δ*dSTE*/Δ*dPTE* in the defined dominant coupling mode supported by a surrogate analysis of 1000 surrogates (*p* < 0.01).

Figure 3 demonstrates an example of time-lag estimation between two ROI time series band-pass-filtered in δ brain rhythm using dSTE.

We also estimated the time lag within and between frequencies across every pair of cortical sources and separately for amplitude and phase domains in four age groups. For a more detailed description of time-delayed information theoretic measures, an interested reader can refer to [76].

### 2.7. Posterior–Anterior Index (PAI) and Posterior–Anterior Time Lag

For each frequency band and subject, the matrices that tabulate the strength of dSTE/dPTE coupling were averaged separately across the 75 epochs, yielding one matrix per subject. These were then averaged across subjects. The average value was subsequently computed for each ROI; that is, the average preferred direction of information flow for a region was also computed. To establish whether there was a consistent pattern of information flow, a posterior–anterior index (PAI) was computed as follows:(17)$$PAI^{\Delta dSTE_{ij}}=\frac{{\left\{\underline{\Delta dSTE_{ij}}\right\}}_{posterior}-{\left\{\underline{\Delta dSTE_{ij}}\right\}}_{anterior}}{{\left\{\underline{\Delta dSTE_{ij}}\right\}}_{posterior}}\;\%\;\;\;$$
(18)$$PAI^{\Delta dPTE_{ij}}=\frac{{\left\{\underline{\Delta dPTE_{ij}}\right\}}_{posterior}-{\left\{\underline{\Delta dPTE_{ij}}\right\}}_{anterior}}{{\left\{\underline{\Delta dPTE_{ij}}\right\}}_{posterior}}\;\%$$
where the Δ*dSTE*/Δ*dPTE* was averaged over a set of posterior and anterior regions, respectively. A positive (%) PAI indicates preferential flow from posterior regions toward anterior regions, and a negative PAI (%) indicates preferential flow from anterior regions toward posterior regions. PAI was ultimately normalized by the maximum observed value within each ROI.

The significance of the PAI was assessed using randomization testing, whereby the average values were permuted across the ROIs, after which the PAI was computed. This was repeated 1000 times to build a distribution of surrogate PAI values against which the observed PAI was tested (*p* < 0.01).

The whole approach was adopted independently for each frequency band and amplitude/phase dynamics.

In the same manner, as for PAI, we estimated the time lag within the studied frequency bands between posterior and anterior brain areas and independently for amplitude and phase dynamics. Statistical levels of the observations were reached via a similar randomization testing approach as that described above for PAI.

### 2.8. Age-Dependent Time Delays within and between Frequencies

We estimated the group-averaged time delays for both within and between frequen-cies. We first averaged the time delays for every pair of sources across temporal segments and then across the entire set of possible pairs of sources. This procedure gave us one value of time delay per subject and per frequency or cross-frequency pair. The whole pro-cedure was repeated separately for each subject and for amplitude and phase domains. Particularly for cross-frequency pairs and for each modulator frequency, we averaged the time delays across the modulated frequencies. For the δ modulator, we averaged the time delays for its five modulated frequencies—δ–θ, δ–α_1_, δ–α_2_, δ–β, and δ–γ to obtain a representation of the temporal scale of the functionality of each frequency when it modulates the rest of the brain rhythms. Finally, we group-averaged the time delays separately for each age group within and between frequencies, as well as amplitude and phase domains. 

For the statistical test, a Wilcoxon rank sum test was adopted to compare age-dependent time delays between age groups per case (*p* < 0.01, Bonferroni-corrected; *p*’ < *p*/6, where 6 refers to the total number of pairwise comparisons across age groups within each frequency band in both BFC and CFC and in both domains separately). This procedure was followed independently per frequency band, functional interaction (BFC or CFC), and domain (amplitude or phase domain). We also adopted the same statistical test to compare the time delays between BFC and CFC per modulating frequency, per age group, and per domain (*p* < 0.01). Our aim in the second analysis was to find significant differences between BFC and CFC time delays across the modulating frequencies and age groups, as well as in both domains. 

### 2.9. Dominant Coupling Modes Model (DoCM)

To reveal the DoCM independently for amplitude and phase dynamics, we adopted the following surrogate analysis to determine: (a) whether a given coupling strength (dSTE/dPTE) differed from what would be expected by chance alone; and (b) whether a given non-zero value (dSTE/dPTE) indicated coupling that was, at least statistically, non-spurious.

Briefly, in our analysis, we used three levels of statistics: surrogate analysis and p-values for each pair of ROIs in every within-frequency interaction and cross-frequency pair (previously described), Bonferroni correction to detect the DoCM for each pair of ROIs, and, finally, FDR to detect the significant interactions across the network.

For every time epoch, source pair, intra-frequency (6 frequencies), and pair of frequencies (15 frequency pairs), we tested the null hypothesis (H0) that the observed dSTE/dPTE value was derived from the same distribution as the distribution of surrogate dSTE/dPTE values. A total of 1,000 surrogate time series were generated independently for each of the 6 + 15 = 21 cases. For each data set, the surrogates of dSTE/dPTE, called dSTEs/dPTEs, were computed. We then determined a one-sided *p*-value expressing the likelihood that the observed dSTE/dPTE value could belong to the surrogate distribution and corresponded to the proportion of ‘surrogate’ dSTEs/dPTEs, which was higher than the observed dSTE/dPTE value [77]. dSTE/dPTE values associated with statistically significant *p*-values were considered unlikely to reflect signals not entailing dSTE/dPTE coupling. Then, we applied a Bonferroni correction to detect (*p*’ < *p*/21) the DoCM per pair of ROIs at every epoch in both estimators.

Three different scenarios are possible in the process of identifying prominent dSTE/dPTE coupling modes associated with a particular pair of source time series and a specific epoch: (A) where only one DoCM (either intra or inter) met this criterion. (B) In the case of two DoCMs, both exceeding the statistical threshold, the one with the highest dSTE/dPTE value was identified as the characteristic dSTE/dPTE coupling mode for this pair of sources in a particular time window (epoch). (C) If none of the intra- or cross-frequency pairs exceeded the statistical threshold, a value of zero was assigned to this pair of sources with no identified characteristic coupling mode.

Then, we applied a false-discovery rate (FDR) method [73] to control for multiple comparisons within every brain network using the p-values derived as the DoCM across all pairs of ROIs. The expected proportion of false positives is set to *q* ≤ 0.01. Finally, the surviving connections expressed the dSTE/dPTE mode that characterized specific pairs of ROIs and was determined based on the highest statistically significant (dSTE/dPTE) value derived from surrogates, Bonferroni correction, and FDR.

For each participant, the resulting time-varying (TV) TV^dSTE^/TV^dPTE^ profiles constituted a 4D array of size [21 (frequencies + pairs of frequencies) × 75 (time windows—epochs) × 78 (sources) × 78 (sources)] that stored the strength and direction of dSTE/dPTE. The identity of promin-ent intra- or cross-frequency interactions for every pair of sensors at each time window (epoch) was ultimately stored in a second 4D array of size [21 × 75 × 78 × 78] using integers ranging from 1 to 21, e.g., 1 for δ, 2 for θ, …, and 21 for β–γ. In a third array with the same dimensions, we kept the time-lag estimations.

The aforementioned procedure was applied independently for amplitude and phase dynamics, leading to the construction of 2 (dSTE/dPTE) × 3 (strength-DoCM time lags) 4D arrays per subject.

Based on the appropriate surrogate analysis and statistical filtering of spurious interactions, we estimated the probability distribution of DoCMs independently for amplitude and phase dynamics. We enumerated the number of DoCMs for each of the 21 cases (intra- and inter-frequency couplings) across the 75 epochs and every possible pair of sources. Afterward, we normalized each of the 21 estimations by their sum to obtain probability distributions of DoCMs across time and the cortex. The aforementioned procedure was applied independently for each subject, epoch, and amplitude/phase dynamics across the interactions of 78 (sources) × 78 (sources).

Probability distributions (PDs) of DoCM can be tabulated in a matrix of 6 × 6 dimensions, where in the in-diagonal, the PDs of the six intra-frequency couplings are inserted, whereas in the off-diagonal, the 21 PDs of the cross-frequency pairs are kept. This matrix is called a comodulogram. For each subject, we estimated the epoch-averaged comodulograms representative of both amplitude and phase dynamics.

### 2.10. Brain Age Index (BAI)

We defined a novel BAI based on the ratio of the sum of PDs of cross-frequency interactions (off-diagonal cells from comodulograms) versus the sum of PDs of intra-frequency couplings (in-diagonal cells from comodulograms) based on the DoCM and estimated over the comodulogram. The proposed frequency-dependent BAI is defined as follows:

A frequency-dependent brain age index (fBAI) can also be defined as:(19)$$fBAI\left(k\right)=\frac{PD\left(k,k\right)}{\frac{1}{\left(N_{modulated}\right)}\sum_{l\neq k}^{Nmodulated}PD\left(k,l\right)}\;$$
where *k* denotes the modulating frequencies {δ, θ, α_1_, α_2_, β}, and *N_modulated_* is the number of modulated frequencies per modulating frequency, e.g., for δ modulating frequency, *N_modulated_* = {θ, α_1_, α_2_, β, γ}; for θ modulating frequency, *N_modulated_* = {α_1_, α_2_, β, γ},...; for β modulating frequency, *N_modulated_* = {γ}.

Using linear, quadratic, Gaussian (centered/non-centered, normalized/non-normalized), exponential, von Bertalanffy with y-intercept, von Bertalanffy, quadratically constrained to the origin and log models, and the coefficient of determination (R), we computed the best model for BAI curves versus real age [78].

### 2.11. Stability of Causal Brain Networks and Time Delay across Time

To assess the similarity of the two functional networks, we estimated the graph diffusion metrics between the original weighted directed effective brain networks [79]. The graph diffusion distance metric (GDD) returns a value from 0 up to a positive value, where 0 means that the two functional brain networks are similar. To access the statistical significance of this similarity between every pair of functional networks, we compared it with random versions of one of the two functional brain networks. Specifically, we created 1000 random functional brain networks by shuffling the directed connections but pre-serving the degree and the strength of each node [80]. From the distribution of 1000 GDD values, we assigned a *p*-value to the original GDD (*p* < 0.01). In addition, the whole proce-dure was repeated between every pair of temporal effective brain networks across the epochs, resulting in 75 × 74/2 = 2775 combinations. Finally, we applied a *z*-score > 2 across the 2775 GDD values to detect outliers, and we kept only the epochs that survived this threshold for further analysis, whereas the relevant GDD deviates from the randomization procedure (*p* < 0.01). The whole approach was repeated separately for each subject, amplitude, and phase domain and for every frequency-based dynamic effective brain net-work, as well as for the effective brain networks based on DoCM. 

It is important to mention here that GDD models the alignment in terms of the infor-mation flow of two effective brain networks by taking into account both the direction and the strength of coupling between two sources. 

To further examine whether connections between sources exhibit consistent latencies across time-resolved effective brain networks, we considered the coefficient of variation (CV) as the mean time delay across temporal segments versus the standard deviations across these blocks. A CV higher than 10 that demonstrates a significant (*p* < 0.01) difference from zero using a one-tailed t-test is acceptable. 

All our results were based on the summary of our evidence that overcomes both statistical requirements described previously based on GDD and CV estimates. Finally, we rejected epochs that did not pass both the whole-brain GDD criterion and the ROI-based CV criterion for more than 10% of the potential pairs of ROIs (78 × 77/2 = 3003 total number of pairs). 

### 2.12. Test–Retest Reliability of Estimates in Repeat MEG Scans

We should underline here that the participants in the large cohort were scanned with the ELEKTA MEG system, whereas the second smallest repeat MEG scan cohort participants were scanned with the MEG CTF system. The supplementary data sets of ten healthy young adults (five women aged 24.4 ± 1.5 years) recorded twice were used to assess the reproducibility of our estimates, as well as to validate brain charts based on BAI. The subjects were scanned at the CUBRIC Neuroimaging centre [81]. Τhe subjects were recorded under a resting-state eyes-closed condition compared to the eyes-open condition of the big cohort. Below, we describe the statistical tests, followed by the repeat-scan cohort. 

We first repeated the same statistical analysis as that described for the first large cohort. We applied a Wilcoxon rank sum test between the two sets of GDD values corresponding to the first and second sessions (*p* < 0.01). We repeated this analysis independently per subject, frequency band, amplitude, and phase domain, as well as for the effective brain networks based on DoCM. 

For the second test, we first estimated the CV of time delay across temporal segments per pair of sources, as in the original cohort. A CV higher than 10 that also demonstrates a significant (*p* < 0.01) difference from zero using a one-tailed *t*-test is acceptable. We also computed the p-value between the two sets of CVs from every possible pair of ROIs (78 × 77/2 = 3003 total number of pairs) linked to the first and second sessions. We applied this procedure for every subject, frequency, amplitude, and phase domain, as well as for the effective brain networks based on DoCM. To this end, we used the Wilcoxon rank sum Test with *p* < 0.01. 

All our results were based on summary of our evidence that overcomes both statisti-cal requirements described previously based on GDD and CV estimates. Finally, we re-jected epochs that did not satisfy both the whole-brain GDD criterion and the ROI-based CV criterion in more than 10% of the potential pairs of ROIs (78 × 77/2 = 3003 total number of pairs) in both sessions. Additionally, we considered findings only from cases (subjects, frequencies, amplitude and phase domains, and DoCMs) where there was no statistical difference between the two sessions. 

The third reproducibility test was assessed by comparing how close the BAI for each new subject was to the fitted curve for the main analysis. To that end, we adapted the Euclidean distance between the estimated BAI and the fitted curve. The BAI for each sub-ject in both amplitude and phase domains was averaged between the two scan sessions only if their Euclidean distance was less than 0.02. Otherwise, these subjects were ex-cluded from the cohort in terms of BAI. 

### 2.13. Computational Effort

The computational effort required to obtain these estimations is massive. In total, we had to obtain the DoCM for every pair of brain regions (3003), estimating the original values of both BFC and CFC (6 + 15 = 21 in total) and the 10,000 surrogates per type of interaction across 75 epochs. This procedure was applied in both estimators and for the 103 subjects. For one subject, the computations are 3003 (pairs of brain regions) × 21 (BFC+CFC) × 75 (epochs) × 2 (estimators) = 9,459,450 estimations of original dSTE + dPE values using both estimators and 1,000 surrogates for each one. To reduce the computational effort and the computational time needed (weeks) to obtain the results in this large database, we ran the whole analysis in a cluster with 100 cores (CUBRIC) using shell scripts in a Linux environment.

### 2.14. Implementation of dSTE and dPTE

The MATLAB implementation of dSTE/dPTE can be found at the following links: (https://github.com/stdimitr/Symbolic_Transfer_Entropy and https://github.com/stdi-mitr/Phase_Transfer_Entropy). The Python implementation of both estimators can be found on the GitHub website of our Dyconnmap module (https://github.com/makism/dyconnmap [82]). 

## 3. Results

### 3.1. Report on Stability of Causal Brain Networks and Time Delay across Time

Based on the brain-based GDD criterion and the ROI-based CV criterion, we rejected 9.86 ± 2.14 epochs averaged across subjects, frequencies, amplitude, and phase domains and 8.17 ± 2.03 epochs in the effective brain networks based on DoCM. Based on these findings, we did not reject any subject across any case (frequency, amplitude, and phase domain and in the DoCM approach). In the following sections, we describe our stable findings in detail.

### 3.2. Dominant Frequency-Dependent Information Flow

We observed a consistent posterior-to-anterior information flow of the phase dynamics in {α_1_, α_2_, β, γ} across temporal segments and an opposite pattern of anterior-to-posterior flow in θ, whereas concerning amplitude dynamics, a posterior-to-anterior information flow in {α_1_, α_2_, γ}, a sensory-motor β-oriented pattern, and anterior-to-posterior pattern in {δ, θ} were revealed. Our results based on dPTE replicated previous findings [33], whereas results based on the amplitude domain are reported here for the first time. It is important to underline that in the δ frequency, an opposite information flow was revealed between amplitude and phase domains. Figure 4 demonstrates the cortical distribution of dSTE/dPTE in the cortex across frequency bands and for both amplitude and phase dynamics, respectively.

### 3.3. Anterior–Posterior Time Lags for Frequency-Dependent Interactions

Time lag within the studied frequency bands was estimated between posterior and anterior brain areas and independently for amplitude and phase dynamics. Based on amplitude dynamics, we detected a positive time lag from posterior to anterior in {α_1_, α_2_, β, γ} and a negative time lag from anterior to posterior in {δ, θ}. Based on phase dynamics, we detected a positive time lag from posterior to anterior in {δ, α_1_, α_2_, β, γ} and a negative time lag from anterior to posterior in θ (Table 2).

The most remarkable evidence was the opposite sign (positive/negative) for δ frequency between phase and amplitude domains, respectively. We rejected three subjects from this analysis as outliers from the whole cohort, although these subjects satisfied the stability of causal brain networks and time delay across time.

### 3.4. Posterior–Anterior Index (PAI)

PAI within the studied frequency bands was estimated between posterior and ante-rior brain areas and independently for amplitude and phase dynamics. Based on ampli-tude dynamics, we detected a positive (%) PAI from posterior to anterior in {α1, α2, β, γ} and a negative (%) PAI from anterior to posterior in {δ, θ}. Based on phase dynamics, we detected a positive PAI from posterior to anterior in {δ, α1, α2, β, γ} and a negative PAI from anterior to posterior in θ (Table 3). The most remarkable evidence was the opposite sign (positive/negative) for δ frequency between phase and amplitude domains, respec-tively. We also rejected three subjects from this analysis as outliers from the whole cohort, although these subjects satisfied the stability of causal brain networks and time delay across time.

### 3.5. Mean Time Delays within and between Frequencies in Amplitude and Phase Domains

In Figure 5, we demonstrate the group-averaged time delays within and between frequency pairs and in both amplitude and phase domains. We clearly detected that in both amplitude and phase domains across groups, the information flow of a specific brain rhythm follows a specific rule both within frequency interactions and between frequencies as a modulator. 

We revealed a significant age group trend for age group 4 (51–60 years), which demonstrates higher mean time delays in cross-frequency interactions in both amplitude and phase domains (Figure 5B,D), in θ–θ phase-to-phase interactions (Figure 5C), and no significant differences in amplitude domain for the within-amplitude interactions (Figure 5A). We also revealed that averaged time delays of all the studied frequency modulators {δ, θ, α_1_, α_2_, β} are significantly lower in cross-frequency interactions compared to within-frequency interactions in both amplitude and phase domains and in the four age groups (comparing Figure 4A vs. Figure 4B and Figure 4C vs. Figure 4D).

### 3.6. DoCM for Amplitude and Phase Dynamics

The main contribution of cross-frequency interactions in the DoCM was from {δ, θ, α_1_} frequencies in both amplitude and phase dynamics. Figure 6 demonstrates the group-averaged comodulograms from amplitude and phase-based DoCM. The modulating frequency is plotted on the horizontal axis, whereas the modulated frequency is plotted on the *y*-axis. The total sum of the PD in the comodulogram equals one.

### 3.7. BAI for Amplitude and Phase Dynamics

Based on comodulograms and the ratio of inter- versus intra-frequency interactions (Figure 6), we defined a BAI that demonstrated sensitivity across age groups (Figure 7). Using linear, quadratic, Gaussian (centered/non-centered, normalized/non-normalized), exponential, von Bertalanffy with y-intercept, von Bertalanffy, quadratic constrained to the origin and log models, and the coefficient of determination (R), we computed the best model for BAI curves versus real age [78]. A single function could not fit the data, but a set of two functions fitting the data separately below and above 30 years of age fitted well.

Finally, we detected that a non-centered, non-normalized Gaussian model fits better to both age-dependent BAI curves in both amplitude and phase domains [83]. The following equation describes the Gaussian fit model with three free parameters (α, σ, and xo):(20)$$y\left(x\right)=a\times exp^{\left(\frac{-{\left(x-xo\right)}^{2}}{2\;x\;\sigma^{2}}\right)}$$

Figure 7 illustrates the proposed BAI tailored to each frequency band and amplitude/phase domain with the fitted Gaussian models in both segments of the curve. Table 4 lists the three free parameters for each frequency band, domain, and curve segment for the whole cohort.

### 3.8. Reliability of Estimations in the Repeat-Scan Cohort (Talk about GDD)

Based on the brain-based GDD criterion and the ROI-based CV criterion, we rejected 10.64 ± 2.57 epochs from the first scan session and 11.09 ± 2.41 epochs averaged across the second scan session across subjects, frequencies, amplitudes, and phase domains. Similarly, we rejected 8.92 ± 1.97 epochs from the first scan session and 10.24 ± 2.49 epochs from the second scan session in the effective brain networks based on DoCM. Based on these findings, we did not reject any subject across any case (frequency, amplitude, phase domain, and in the DoCM approach). In ‘Section 3.9’, we describe, in detail, our stable findings. No statistical difference was detected between the two scan sessions across any case (subjects, frequencies, amplitude, phase domain, and DoCM). The repeatability of the estimates derived from the repeat-scan cohort supported our analytic plan and the reported information flow across all measurements (strength, direction, time delay, PAI).

### 3.9. Reproducibility of BAI for Amplitude and Phase Dynamics across MEG Systems (ELEKTA-CTF)

It is important to mention that the whole repeat-scan cohort showed highly repeatable BAI in both amplitude and phase domains. Moreover, the Euclidean distance of the session-averaged BAI for every subject was less than 0.02 in both amplitude and phase domains and across the frequency modulators {δ, θ, α_1_, α_2_, β} when it was compared with the brain charts based on the suggested BAI (Figure 7). These significant findings supported the repeatability of the proposed BAI and its stability across MEG systems and validated the brain charts based on BAI.

## 4. Discussion

Using MEG beamformed source-reconstructed activity and proper neuroinformatic tools, including connectivity estimators and statistics, we provided, for the first time, evi-dence in large-scale brain networks that the network topology of effective networks is repeatable across the experimental day and within a one-week rescan session. The current study provides further support for claims that human brain communication is realized via stable pathways that exhibit reliable direction and precise timing [22,23]. 

Our main results can be summarized as followings: 

We confirmed a well-established posterior-to-anterior information flow of the phase dynamics in {α_1_, α_2_, β, γ} and an opposite pattern of anterior-to-posterior information flow in θ, whereas with respect to amplitude dynamics, we detected a posterior-to-anterior information flow in {α_1_, α_2_, γ}, as well as a sensory-motor β-oriented pattern and anterior-to-posterior pattern in {δ, θ}. 

We detected time delays between neuromagnetic source activity within and between frequencies and in both amplitude and phase domains, ranging from approximately 90 ms (δ) to 15 ms (γ). A positive time lag from posterior to anterior in {α_1_, α_2_, β, γ} and a negative time lag from anterior to posterior in {δ, θ} was revealed in the amplitude domain, whereas a positive time lag from posterior to anterior in {δ, α_1_, α_2_, β, γ} and a negative time lag from anterior to posterior in θ was detected in the phase domain. The most striking pattern was the opposite flow of information in the δ band for phase and amplitude. 

We revealed a positive (%) PAI from posterior to anterior in {α_1_, α_2_, β, γ} and a negative (%) PAI from anterior to posterior in {δ, θ} based on amplitude dynamics. Based on phase dynamics, we detected a positive (%) PAI from posterior to anterior in {δ, α_1_, α_2_, β, γ} and a negative (%) PAI from anterior to posterior in θ. 

Age group 4 (51–60 years) demonstrated significantly higher mean time delays in cross-frequency interactions in both amplitude and phase domains and in θ–θ phase-to-phase interactions compared to the rest of the groups.

Group-averaged time delays in cross-frequency interactions in both amplitude and phase dynamics were significantly lower for all the studied frequency modulators compared to the intra-frequency interactions in both amplitude and phase domains.

The main contributors of CFC based on DoCM were {δ, θ, α_1_} frequencies.

Based on DoCM, we defined a novel frequency-dependent BAI that untangles a clear age-dependent trend of the suggested index in both amplitude and phase domains. This BAI can be seen as a maturity index tailored to MEG complementary to the fMRI-based maturation index [83] and structural MRI [84].

Our results were highly repeatable in a repeat-scan cohort, also supporting the reproducibility of the cross-MEG system.

### 4.1. Dominant Information Flow between Posterior and Anterior Cortical Areas

With dSTE, posterior regions of the DMN were found to be senders of information in the high-frequency bands (8−45 Hz) and receivers in the θ band. The DMN is an active brain area at the resting state and has been directly linked to internal mentation and to an unconscious awareness of the external world [85,86]. DMN is formed by two spatially distinct brain systems that interact: the temporal system is involved in memory processes, and the fronto-parietal system is linked to self-referencing mental activity. 

With the support of the dPTE, we revealed a well-established posterior-to-anterior information flow of the phase dynamics in {α_1_, α_2_, β, γ}, an opposite pattern of anterior-to-posterior information flow in θ, a posterior-to-anterior information flow in {α_1_, α_2_, γ}, and an anterior-to-posterior pattern in {δ, θ}, further supporting the formation of a loop of this frequency-dependent sender/receiver brain area [33,87]. 

A dominant posterior-to-anterior pattern of information flow in the high-frequency bands between parieto-occipital and frontal brain areas and a simultaneous anterior-to-posterior pattern from frontal regions to temporal and posterior regions in the θ frequency could support the hypothesis that these subsystems form a loop or an integrated system [87,88]. Information circulates through this system. Evidence that the θ band is the key frequency for memory processes in the frontal and hippocampal areas further supports this interpretation [21,89]. 

Brain connectivity in both α and β frequencies is important to attention, where θ frequency connectivity from the medial frontal cortex to many brain areas plays a key role in inducing control from the higher association brain areas over the lower-level areas and perceptual processes, as well as over the DMN [90] Our findings in the θ frequency are in line with this theory, which explains why we detected distributed θ information flow from frontal brain areas to many brain areas, including the DMN. Simultaneously, the anterior-to-posterior α connectivity supports a gating mechanism for attention due to the top-down modulation by α activity, which inhibits irrelevant activity [90,91]. However, we observed an opposite posterior-to-anterior dominant pattern of information flow in the α frequency, although it is possible that the observed enhanced bottom-up signaling in the α frequency is in itself a consequence of enhanced top-down signaling in the θ frequency [92]. 

We should underline that our study focused on the analysis of resting-state ongoing activity and was not task-based. Moreover, different forms of attention are linked to different spatiotemporal and frequency contents [93]. 

It is important to note that the δ frequency showed an opposite anterior–posterior pattern between amplitude and phase domains. The δ frequency band has been directly linked to learning and reward processes [94], as well as to memory encoding and retrieval [95]. A simultaneous EEG-fMRI study showed that δ source activity is located frontally and mainly in the anterior cingulate cortex [96], whereas it was linked to internal memory retrieval from the past, daily internal thoughts [97], and cognition [98]. A visual percep-tion study revealed that δ activity (2–4 Hz) in the prefrontal cortex tracked task context and modulated sensory processing in a top-down control. This study concluded, via CFC and using a phase-to-amplitude (PAC) coupling estimator, that the δ phase mediated top-down control of posterior α brain amplitude activity for visual perception [99]. Another ECoG study following a PAC analysis untangled a δ/θ phase modulation of high γ activity in sub-second facilitations that coordinate the fronto-parietal cortex, whereas this modulation is guided by attentional demands [100]. The fronto-parietal attention network is a collection of brain areas located in the frontal and parietal cortices and is crucial for the control of attentional selection processes [101,102,103]. 

An important study in oscillatory phase coupling theory showed clear evidence of how neuronal oscillations enable selective, target, and dynamic control of anatomically distributed functional cell assemblies [104]. This observation is supported by phase-cou-pling rates, even between distant cortical brain areas. The findings of this study complement the communication through coherence (CTC) hypothesis suggested by Fries [16], whereby phase differences can modulate the effective connectivity between two cortical areas [105]. The hypothesis described by Canolty et al. [104] postulates that distributed LFP activity influences spiking activity and incorporates N distinct phase signals simultaneously. This hypothesis extends CTC theory in N distinct phase signals. Canolty’s theory showed that spiking activity in single neurons depends on the whole pattern of oscillatory phases occurring in many brain areas and that these phase-coupling patterns have an impact on long-range communication. 

Oscillations play a key role in cognition, perception, and action, which is supported by findings that oscillatory activity is entrained by sensory [106], linguistic [107], and mo-tor events [108]. This entrainment depends on attention [106,108] and provides a link to internal processes critical for memory and learning processes associated with characteristic low-frequency brain rhythms [109]. 

Canolty’s study showed that neurons are sensitive to multiple frequencies [110,111]. The cellular and network origins of distinct brain frequencies are the focus of ongoing research [112]; however, the period of concatenation hypothesis [113] provides a support-ive mechanism accounting for the generation of the frequency bands observed in the ne-ocortex. Each generated distinct brain frequency can be independently controlled by dif-ferent neuronal ensembles. The fluctuation of patterns of oscillatory coupling across mul-tiple anatomically dispersed brain areas coordinates distinct neuronal cell assemblies [104]. Different neuronal assemblies with similar phase-coupling preferences depend on the functional role of neurons and correlate with behavior, suggesting that neuronal os-cillations may synchronize anatomically dispersed ensembles actively engaged in func-tional roles [104]. 

In our study focusing on the analysis of resting-state recordings, we believed that the anterior–posterior patterns across frequencies and in both amplitude phase domains characterized the ongoing activity. The ongoing brain activity can be described by visual attention demands, internal thinking, and memory retrieval, anatomically involving the DMN, the visual network, and the fronto-parietal attentional network, as well as higher association brain areas over the lower-level areas and perceptual processes. 

The δ phase in the primary visual cortex is entrained to the rhythm of a stream of attended stimuli, resulting in increased response rates [106]. The δ amplitude originates from the prefrontal cortex during context-dependent top-down processing [99]. We can assume that the opposite anterior–posterior pattern between amplitude and phase causal patterns detected in the δ frequency could be local and related to distributed functional demands over frontal (amplitude) and parietal (phase) brain areas related to internal thoughts, as well as suppression of irrelevant activity [97]. We also assumed that this interplay exists simultaneously in ongoing activity supported by Canolty’s theory [31]. 

Our analysis was applied separately to amplitude and phase domains to reveal their different roles in basic brain rhythms in the resting state [114,115].

### 4.2. Spatiotemporal Distribution of Time Delays within and between Frequencies

Brain oscillations have a natural logarithmic relationship with each other to support the communication of neuronal networks with different sizes and types of connections to coordinate their activity [116]. It is well known that the temporal window of activation and the activation phase vary in relation to the length of an oscillation period. This means that the related time delays are lower compared to slower brain oscillations. The large repertoire of frequency bands and their logarithmic relationship may serve as a mecha-nism to overcome any information processing limitations due to synaptic delays [116,117]. 

Here, we provide the first demonstration of time delays of coordinated activity be-tween two areas operating within the same and across different frequency bands in each area. It is well known that cross-frequency coupling of brain oscillations is a key mecha-nism in spatiotemporal coordination of brain activity (e.g., [31,118]). The time delay be-tween two or more brain areas that oscillate on a dominant frequency can inform how time shapes brain function or dysfunction and how it is connected to behavior. We revealed a mechanism whereby the time delay between the frontal^θ^-parieto-occipital^α2^ cortices discriminates correctly from wrong calculations in a mental arithmetic task [75].

We demonstrated that the mean time delays of the modulators in cross-frequency interactions are significantly shorter compared to within-frequency interactions in both amplitude and phase domains. This further supports the reason why cross-frequency in-teractions are important for the fast and accurate exchange of information between remote brain areas. 

Brain activity is inherently rhythmic and anatomically distinct and spans several tem-poral scales. The concept of CFC has been proposed as one solution for information inte-gration across several spatiotemporal scales [31,99]. These findings suggest that the brain uses both frequencies- and time-division multiplexity to optimize directional information transfer. 

### 4.3. Dominant Intrinsic Coupling Modes (DoCMs)

We independently detected DoCMs for intra- and cross-frequency coupling for both amplitude and phase dynamics using the dSTE and dPTE ([34,49,75]). We revealed a complementary pattern of the DoCM in both domains with main contributions for cross-frequency coupling in δ, θ, and α_1_ frequencies. The main contributors to intra-frequency bands were δ, α_2_, and β [87].

### 4.4. BAI Based on DoCM

The functional role of cross-frequency coupling has been studied across different tar-geted groups [31,46,47,48]. To the best of our knowledge, this is the first study to explore cross-frequency interactions for a range of ages in the source space. A recent study re-ported inefficient communication of the default mode network brain areas based on cross-frequency couplings linked to age-related short-term memory decline [119]. Another study using source-reconstructed MEG resting-state activity and following a static func-tional brain connectivity approach reported a maturation index based on the global strength of the coupling, revealing an asymptotic curve for γ frequency until the age of 29 years and a linear curve for β frequency that did not asymptote, even in adulthood [120]. This study differed from ours in at least five important ways. First, the analysis did not take into account directionality and time lag but was based on undirected functional con-nectivity. Second, the analysis focused only on within-frequency band coupling, ignoring cross-frequency coupling. Third, the age range of the cohort was < 29 years, an age where our analysis extended to 60 years, showing a change in the overall pattern. Fourth, we analyzed amplitude and phase dynamics separately. Finally, our BAI based on DoCM works within the multiplex patterns of amplitude and phase human brain dynamics. 

Taking the ratio of both intra- (BFC) and inter-frequency coupling modes (CFC), our BAI definition is independent of linear dependence and of absolute magnitudes. The results reveal to the best of our knowledge, for the first time—an age trajectory from the end of adolescence (~18 years of age) to the beginning of old age (~60 years of age) that is similar across frequencies and similar for the amplitude and phase domains. This trajectory is also intuitively appealing, as it conforms to our understanding of aging over the range of 18 to 60 years, i.e., a fast climb from adolescence to adulthood, reaching a peak between 20 and 30, followed by a slow decline. An apparent instability around 50 to 60, settling to a slower decline in the last five years of the range, may be due to mental decline becoming more obvious in this age range, leading to the exclusion from the sample of people showing symptoms of decline.

Our results were cross-MEG system reproducible, which further supports the con-sistency of our results. Additionally, our estimates, including both amplitude and phase domains, were highly reliable in the repeat MEG scan cohort. No reference normal stand-ards exist in functional neuroimaging to track the individual differences across ages in a similar way to growth charts for normal height and weight. In our previous work, we adopted a dynamic functional connectivity approach to determine DoCM in resting-state neuromagnetic recordings using complementary undirected connectivity measures [121]. The whole analysis was realized in the sensor space, revealing an inverse U-shaped curve among healthy participants for a measure called brain flexibility. The importance of the lifespan brain chart (8–60 years) based on brain flexibility was further evaluated with re-peat scans in cross-MEG systems, as well as in two cohorts: a dyslexic and a mild trau-matic brain injury group [121]. Our present and previous studies are the first functional neuroimaging studies in the literature that attempt to map important attributes of func-tional/effective connectivity across the lifespan. A recent multi-cohort MRI study pro-vided a standardized measure of atypical brain structure based on MRIs from tens of cross-sectional studies, revealing potential deviations from normal neuroanatomical var-iation in targeted neurological and psychiatric disorders [84].

### 4.5. Multiplexity of Brain Oscillations under Information Flow and DoCM

Our study underlines the importance of adopting multiple estimators to investigate the fluctuation of effective connectivity patterns in both BFC and CFC. The coupling of neuronal assemblies with similar amplitude or phase-coupling preferences, even at long distances, is modulated by their functional role and correlates with behavior [104]. This observation suggests that neuronal oscillations via BFC or CFC scenario may synchronize anatomically and functionally distant neuronal assemblies that are engaged in specific functional roles. The concept of CFC has been proposed as a means of information inte-gration across multiple spatiotemporal scales [31]. Neuronal oscillations play a significant role in coordinating functionally distinct neuronal assemblies that are responsible for communication in large-scale brain networks [31]. To investigate the multiplex character of communication between brain areas on the macroscale, we adopted two estimators that can capture the strength, time delay, and direction between ROI and frequency-dependent brain activity. Moreover, our DoCM revealed the dominant coupling mode per pair of brain areas based on the hypothesis that if two brain areas exchange information, this should be characterized by a specific coupling mode [7,12]. The DoCM model revealed an important BAI, which could be used complementary to structural MRI and normal brain charts [84]. 

## 5. Conclusions

The proposed spatiotemporal investigation of the direction, strength, and time delays of effective coupling within and between frequencies in both amplitude and phase domains is suitable for indexing the development, maturation, and slow decay of the human brain from neonates through adolescence, adulthood, and old age. We reported the first thorough analysis covering the age range from the end of adolescence to the beginning of old age. It is important to continue this work in three directions. First, we need to study the earlier period from birth through childhood and adolescence, when there are periods of tremendous change in synaptic density, myelination, and maturation, especially in the frontal lobes. Second, the aging process should come under scrutiny, especially in a period of our civilization when the proportion of aged people is increasing. Finally, for each specific age group, clinical populations should be studied, augmenting our definitions to include BAI definitions with regional dependence in an effort to elucidate mechanisms of pathology.

## Figures and Tables

**Figure 1 brainsci-12-01404-f001:**
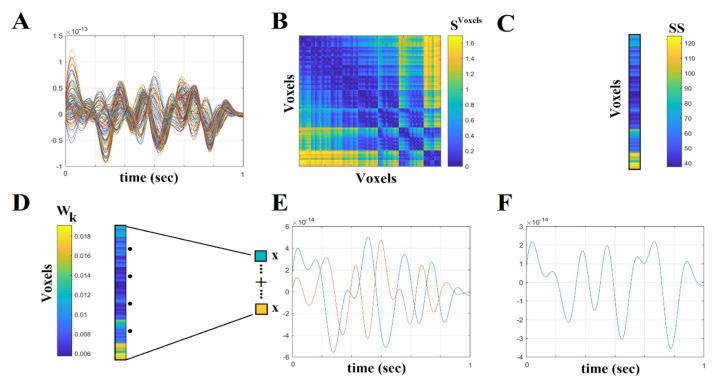
Step-by-step construction of the representative virtual sensor time series for each ROI. (**A**) The plot of the 108 voxel time series falls within the left precentral gyrus ROI. (**B**): Distance correlation matrix $S^{Voxels}$ derived by the pairwise estimation of the 108 voxel time series. (**C**) Summation of the columns of $S^{Voxels}$ produced the vector SS. (**D**) Normalisation of vector SS, which further produces *W_k_*, where its sum equals one. (**E**) Multiplication of every voxel time series by the related weight from the *W_k_*. In this example, we demonstrated this multiplication for the first and last voxel time series. (**F**) The estimated time series for left precentral gyrus activation ((*ROI^activity^*) was obtained by summing the weighted versions of every voxel time series (as in (**E**)).

**Figure 2 brainsci-12-01404-f002:**
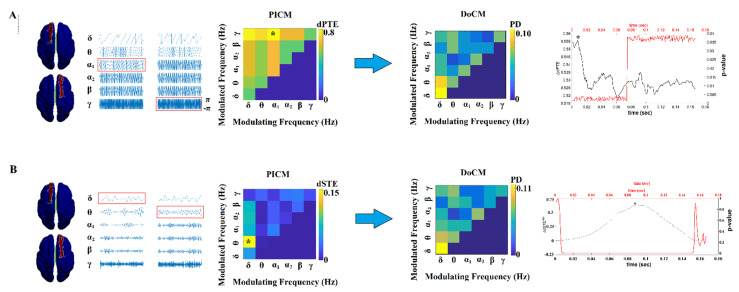
Schematic illustration of the approach employed to identify the dominant coupling mode between two AAL atlas ROIs (left superior frontal gyrus and right superior frontal gyrus) for an epoch (t_1_) during a resting-state MEG recording. In this example, the causal interdependence between band-passed signals from the two virtual sensors was indexed by (**A**) dPTE and (**B**) dSTE. In this manner, both dPTE and dSTE were computed between the activity of the two virtual sensors either for same-frequency oscillations (within-frequency coupling, e.g., δ to δ) or between different frequencies (cross-frequency coupling, e.g., δ to θ). All 21 estimations (within-frequency (6) + cross-frequency coupling (15)) are called potential intrinsic coupling modes (PICMs). PICMs are tabulated in a matrix of size frequencies × frequencies, wherein the main diagonal, the within-frequency couplings are stored, whereas in the off-diagonal, the cross-frequency couplings are tabulated. To reveal the dominant coupling modes (DoCM) per pair of virtual sensors at every epoch, we adopted surrogate data analysis to create a reference and assess whether original dPTE and dSTE values were statistically different from chance. For this example, we revealed α_1_-γ and δ-θ as dominant interactions (DoCM) for dPTE and dSTE, respectively. Following the same procedure across every pair of sensors, we ultimately estimated the probability distribution (PD) of DoCM for both estimators across the 90 virtual sensors. dPTE and dSTE revealed that the first ROI drives the second ROI in both the phase and amplitude domain, whereas the time delay was 10 ms and 86 ms, respectively, as indicated by ‘*’.

**Figure 3 brainsci-12-01404-f003:**
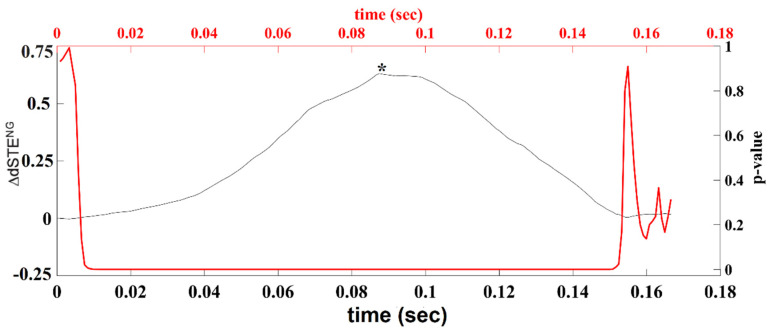
Time-lag estimation between the left precentral gyrus and left superior parietal lobule time series filtered in the δ-band frequency. The example is from a 20-year-old male subject. The black time series illustrates the ΔdSTE^NG^ values for each ms in the displayed 167 ms range. The red time series demonstrates the *p*-values computed for each time lag (ms) after applying surrogate analysis. The *p*-values surrounding the maximum ΔdSTE^NG^ are significant (*p* < 0.01). Finally, we selected the maximum and significant value of ΔdSTE^NG^, which is 1.33 at the time lag of 0.08 s. The horizontal red line extends over the significant time lags around that with maximum ΔdSTE^NG^ value. The left *x*-axis corresponds to the strength (ΔdSTE^NG^) of the coupling, whereas the right *x*-axis (**red**) corresponds to the *p*-values.

**Figure 4 brainsci-12-01404-f004:**
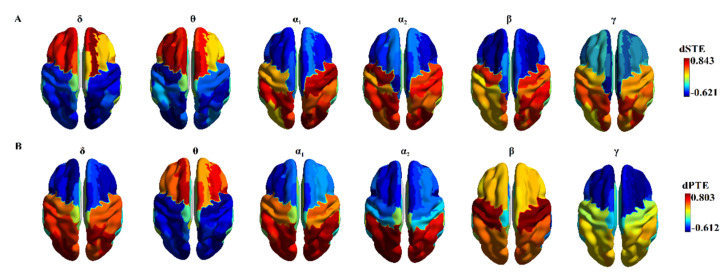
The mean dSTE for each ROI is displayed as a color map across the cortical areas for (**A**) mean amplitude and (**B**) mean dPTE for each ROI displayed as a color phase-oriented information flow. These topographies were stable across temporal segments, revealing a posterior-to-anterior information flow of the phase dynamics in {α_1_, α_2_, β, γ} and an opposite pattern of anterior-to-posterior information flow in θ. Concerning amplitude dynamics, a posterior-to-anterior information flow in {α_1_, α_2_, γ}, a sensory-motor β-oriented pattern, and an anterior-to-posterior pattern in {δ, θ} were revealed.

**Figure 5 brainsci-12-01404-f005:**
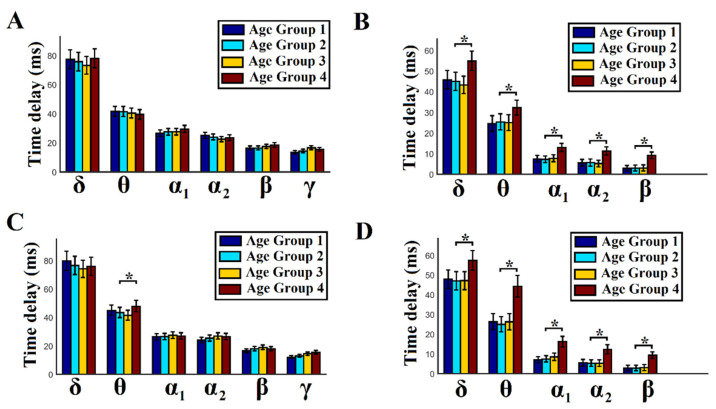
Group-averaged time delays within and between frequencies. (**A**) Group-averaged time delays within frequencies in the amplitude domain. (**B**) Group-averaged time delays between frequencies in the amplitude domain. (**C**) Group-averaged time delays within frequencies in the phase domain. (**D**) Group-averaged time delays between frequencies in the phase domain. * denotes that age group 4 is significantly different from all other age groups. (Age group 1: 18–27 years, age group 2: 28–40 years, age group 3: 41–50 years, and age group 4: 51–60 years). For the statistical test, a Wilcoxon rank sum test was adapted (‘*’ *p* < 0.01, Bonferroni-corrected; *p*’< *p*/6, where 6 refers to the total number of pairwise comparisons across age groups within each frequency band in both BFC and CFC, as well as in both domains separately).

**Figure 6 brainsci-12-01404-f006:**
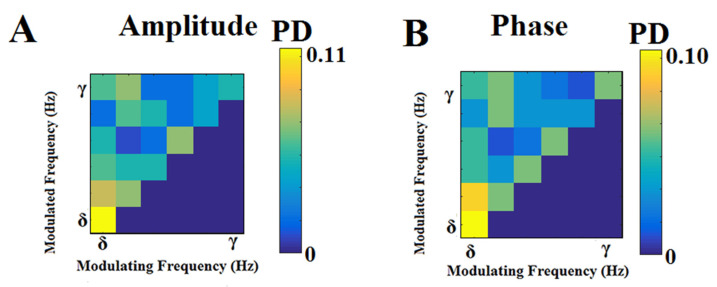
Group-averaged probability distributions (PD) of dominant intrinsic coupling modes (DICMs) based on (**A**) amplitude and (**B**) phase dynamics. The modulating frequency is plotted on the horizontal axis, whereas the modulated frequency is plotted on the *y*-axis. The total sum of the PD in the comodulogram equals one.

**Figure 7 brainsci-12-01404-f007:**
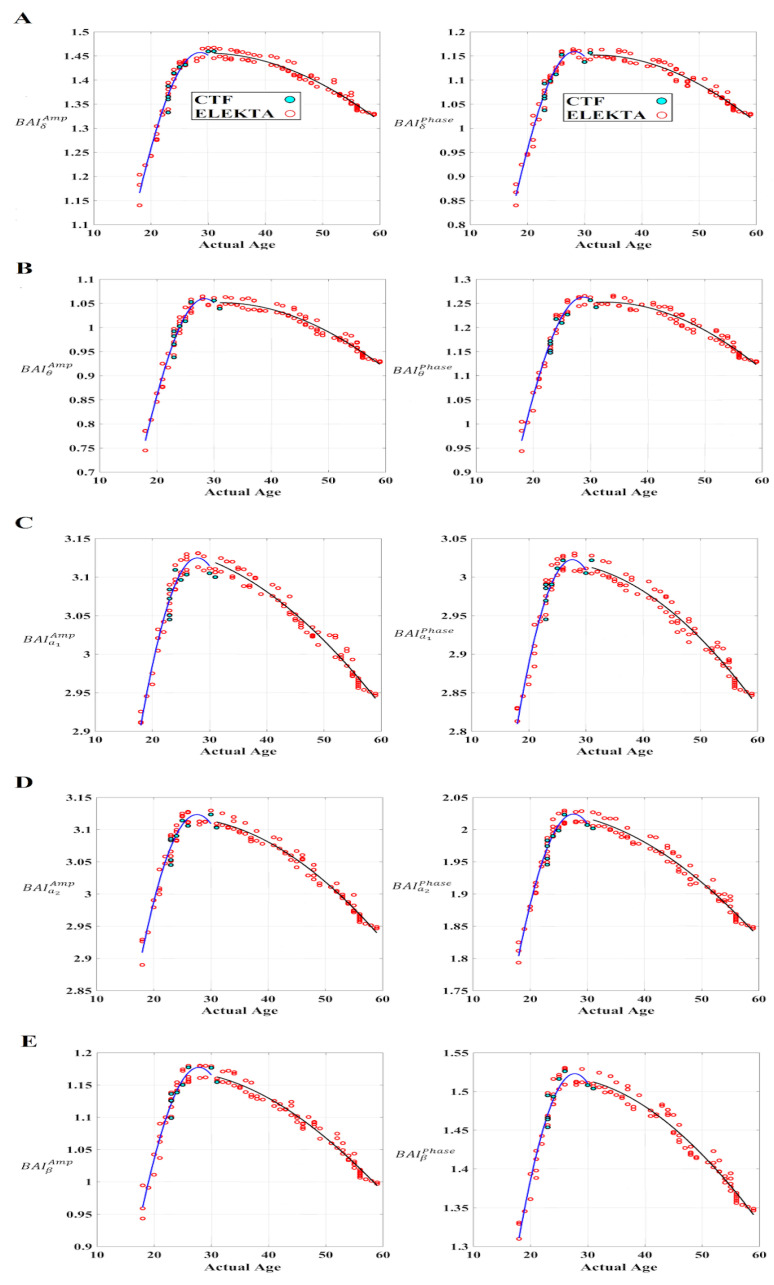
The proposed frequency-dependent brain age index (BAI) is plotted versus actual age. (**A**–**E**) Frequency-dependent BAIs are independently plotted versus actual age for each modulating frequency from δ to β. The first column refers to amplitude-based BAIs, and the second column refers to phase-based BAIs. For the model selection of BAI curves versus real age, we utilized the coefficient of determination (R). The Gaussian model fits better to the two age-dependent BAI curves (**blue color**: 18–30/**black color:** 31–60). CTF/ELEKTA MEG systems refer to the original lifespan cohort and the repeat scan cohort, respectively.

**Table 1 brainsci-12-01404-t001:** Distribution of the 103 participants across the four age groups by gender.

Age group (years)	18–27	28–40	41–50	51–60
*n*	30	25	23	25
Males/females	13/17	11/14	14/9	14/11

**Table 2 brainsci-12-01404-t002:** Subject-averaged PAI^time-lag^ estimation between posterior–anterior cortical areas based on dSTE.

Time Lag(ms)	δ	θ	α_1_	α_2_	β	γ
**Phase**	**76.06** ± 5.61 *	−**40.74** ± 3.48 *****	**27.75** ± 2.91 *****	**23.77** ± 2.95 *****	**17.23** ± 2.51 *****	**15.01** ± 1.78 *****
**Amplitude**	−**76.61** ± 6.17 *****	−**44.48** ± 5.62 *****	**26.85** ± 3.01 *****	**25.75** ± **3.12 ***	**17.86** ± 2.93 *****	**13.75** ± 1.39 *****

* *p* < 0.01; three subjects were excluded as outliers from the whole cohort.

**Table 3 brainsci-12-01404-t003:** Posterior–anterior index (PAI) for amplitude and phase dynamics.

PAI	δ	θ	α_1_	α_2_	β	γ
**Phase**	**15.81** ± 2.34 **% ***	−**16.01** ± 2.11**% ***	**17.17** ± 2.01 **% ***	**14.11** ± 2.31 **% ***	**17.12** ± 2.42 **% ***	**14.76** ± 1.93 **% ***
**Amplitude**	−**16.34** ± **2.72% ***	−**14.85** ± **1.76% ***	**15.91** ± 2.12 **% ***	**18.69** ± **3.11% ***	**18.21** ± 2.77 **% ***	**16.19** ± 2.41 **% ***

* *p* < 0.01; three subjects were excluded as outliers from the whole cohort.

**Table 4 brainsci-12-01404-t004:** Parameters of the non-centred, non-normalised Gaussian fit model to BAI across frequency bands for the whole cohort for amplitude and phase domain.

4A Amplitude Domain				
	α (1st–2nd Curve)	σ (1st–2nd Curve)	xo (1st–2nd Curve)	R (1st–2nd Curve)
${\mathrm{BAI}}_{\delta }^{\mathrm{Amp}}$	1.45–1.45	15.8–67.3	28.5–29.5	0.97–0.95
${\mathrm{BAI}}_{\theta }^{\mathrm{Amp}}$	1.06–10.5	12.82–54.57	28.35–31.20	0.97–0.95
${\mathrm{BAI}}_{\alpha 1}^{\mathrm{Amp}}$	3.12–3.13	26.01–114.44	27.87–17.82	0.97–0.97
${\mathrm{BAI}}_{\alpha 2}^{\mathrm{Amp}}$	3.12–3.11	25.38–101.68	27.57–23.98	0.96–0.97
${\mathrm{BAI}}_{\beta }^{\mathrm{Amp}}$	1.17–1.17	15.37–62.72	27.81–22.93	0.95–0.97
**4B Phase Domain**				
	**α (1st–2nd Curve)**	**σ (1st–2nd Curve)**	**xo (1st–2nd Curve)**	
${\mathrm{BAI}}_{\delta }^{\mathrm{Phase}}$	1.15–1.15	13.23–55.7	28.22–31.7	
${\mathrm{BAI}}_{\theta }^{\mathrm{Phase}}$	1.26–1.25	14.67–56.75	28.77–32.46	
${\mathrm{BAI}}_{\alpha 1}^{\mathrm{Phase}}$	3.02–3.01	25.15–100.31	27.62–24.13	
${\mathrm{BAI}}_{\alpha 2}^{\mathrm{Phase}}$	2.02–2.02	19.98–86.19	27.60–21.20	
${\mathrm{BAI}}_{\beta }^{\mathrm{Phase}}$	1.52–1.51	17.77–69.84	27.73–24.05	

## Data Availability

The resting-state MEG data from the Open Access Omega Project can be downloaded from the pro-ject’s website https://www.mcgill.ca/bic/neuroinformatics/omega.

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
