# Peer review of "Universal Lifespan Trajectories of Source-Space Information Flow Extracted from Resting-State MEG Data"

_brainsci, 2022, doi:10.3390/brainsci12101404_

Round 1

Reviewer 1 Report

In this study, the author explored the information flow in resting-state MEG signals and examined the lifespan trajectory of such flow. The results are clearly presented, though, in my opinion, the methodology and discussion can be improved for better understanding.

Major comments:

1)      The author provided a very detailed, step-by-step description of the methodology adopted in this study, which can be helpful for replicating this study or applying the same method in another study. However, a reader may sometimes lose the main point when being presented with a lengthy description of the method. What may help is to provide a summary of the methods before diving into the equations, e.g. a few paragraphs supplemented with a schematic diagram. With a bigger picture given upfront, the readers may find it easier to understand why the author chose to do a particular calculation and follow the flow of logic.

2)      Related to the first point, since the methodology adopted in this study is complicated, it bears the risk of being “over-engineered” if little discussion around the neural biology or neural mechanism is provided to associate the calculated measures with what actually happens in the brain. The numbers derived from this method are more meaningful if they can be associated with some known facts / previous findings of the brain. This is briefly mentioned in section 4.1, but I suggest adding more in the discussion.

3)      The Introduction does not cover all necessary background information to understand this paper. For example, the background of DoCM, ICM, dSTE and PE is very brief and not exactly to the point. The author is expected to at least cover what they are, how they are calculated (in lay language and brief), how previous studies applied these methods, and what the findings are. The author may consider move some parts of the Method to the Introduction. The goal is to make it clear to the readers about why this measure is used, and what we can expect to know from this measure.

4)      Method 2.9 has multiple steps and is difficult to follow. Consider adding a schematic diagram for this subsection to make it clear to the readers.

5)      In Discussion, the author mentioned “the opposite sign for theta frequency between phase and amplitude supports its distinct functional role in both domain”. As this is one of the major findings for this part of analysis (as noted by the author in Results), the author needs to add more elaborations to clearly explain what it means.

Minor comments:

1)      In-text citation style should be consistent.

2)      Figure 2: Where is the “horizontal bar” mentioned in the caption? Also, just to be clear, the asterisk denotes “p<0.1” for significance?

3)      The equations and in-text maths notations have font size issues (sometimes too big, sometimes too small).

4)      Line 369: this was repeated (10,000) times

5)      Figure 4: show one legend only if they are identical. Replace “age group x” in the legend with the actual age range for each group. Also, explain what “*” means in captions (e.g., * denotes age group 4 is significantly different from xxx)

6)      Show Figure 6 in its original ratio. Also, the equations in this figure are not intelligible.

Author Response

In this study, the author explored the information flow in resting-state MEG signals and examined the lifespan trajectory of such flow. The results are clearly presented, though, in my opinion, the methodology and discussion can be improved for better understanding.

Answer: We would like to thank you for your comments and the chance to review our study. We revised most parts of the study and the changes in the text are underlined in blue color.

Major comments:

  • The author provided a very detailed, step-by-step description of the methodology adopted in this study, which can be helpful for replicating this study or applying the same method in another study. However, a reader may sometimes lose the main point when being presented with a lengthy description of the method. What may help is to provide a summary of the methods before diving into the equations, e.g. a few paragraphs supplemented with a schematic diagram. With a bigger picture given upfront, the readers may find it easier to understand why the author chose to do a particular calculation and follow the flow of logic.

Answer: You are right. We added to section 2.4 with a schematic diagram in order to explain the core of the methodology without using any equation. We hope that every reader can have a clear view of our approach before reading the details of the methodology.

  • Related to the first point, since the methodology adopted in this study is complicated, it bears the risk of being “over-engineered” if the little discussion around the neural biology of neural mechanism is provided to associate the calculated measures with what actually happens in the brain. The numbers derived from this method are more meaningful if they can be associated with some known facts / previous findings of the brain. This is briefly mentioned in section 4.1, but I suggest adding more in the discussion.

Answer: You are right. We attempted to interpret our findings here with known facts. You should consider that our study is unique in many factors and there is no such study in the literature. The majority of studies explored only causal interactions in within-frequencies and also in many cases between targeted predefined brain regions instead of a whole-brain exploratory analysis.

Our study is an empirical and exploratory unique study that explores causal interactions both between and within frequencies for the very first time in the literature. A previous study [Hillebrand et al., 2016] explored causal interactions using only PTE and only in within-frequency couplings. Both this study and ours are the only studies that explore causal interactions across the whole brain in general. There is no mechanistic/modeling study that can further establish a neurophysical model that can take all the parameters explored here into account. For that reason, BAI, time delays, and the probability distribution of DoCM should be interpreted with current knowledge in large-scale brain networks.

Moreover, our study focused on MEG resting-state recordings. It is important to follow a similar analysis of MEG recordings from individuals while performing a task and also in specific brain diseases to further understand the importance of these measures and how specific groups deviate from the normal trajectories.

  • The Introduction does not cover all the necessary background information to understand this paper. For example, the background of DoCM, ICM, dSTE and PE is very brief and not exactly to the point. The author is expected to at least cover what they are, how they are calculated (in lay language and brief), how previous studies applied these methods, and what the findings are. The author may consider moving some parts of the Method to the Introduction. The goal is to make it clear to the readers why this measure is used, and what we can expect to know from this measure.

Answer: Thank you for your comment. We revised the introduction by adding more information to our method. The adding information is coded with blue color.

  • Method 2.9 has multiple steps and is difficult to follow. Consider adding a schematic diagram for this subsection to make it clear to the readers.

Answer: You are right. We added section 2.4 with a schematic diagram in order to explain how the DoCM is performed. The reader can understand the methodology without the need of reading our previous studies.

  • In the Discussion, the author mentioned, that “the opposite sign for theta frequency between phase and amplitude supports its distinct functional role in both domains”. As this is one of the major findings for this part of the analysis (as noted by the author in Results), the author needs to add more elaborations to clearly explain what it means.

Answer: We extended the interpretation of our findings based on major findings in the neuronal communication theory.

Minor comments:

  • In-text citation style should be consistent.

Answer: We passed the whole text in order to be consistent in style.

  • Figure 2:
  1. Where is the “horizontal bar” mentioned in the caption?
  2. Also, just to be clear, the asterisk denotes “p<0.1” for significance?

Answer: Thank you for these important comments.

  1. We corrected it as ‘The horizontal red line
  2. The p-value should be p < 0.01 (corrected).

  • The equations and in-text maths notations have font size issues (sometimes too big, sometimes too small).

Answer: We rewrote the math equations.

  • Line 369: this was repeated (10,000) times

Answer: Done.

  • Figure 4: show one legend only if they are identical. Replace “age group x” in the legend with the actual age range for each group. Also, explain what “*” means in captions (e.g., * denotes age group 4 is significantly different from xxx)

Answer: Very important comment. We added the statement related to the meaning of ‘*’ while we added in the caption the actual age range per group.

  • Show Figure 6 in its original ratio. Also, the equations in this figure are not intelligible.

Answer: We removed the equations from figure 6. The plots of BAI are in their original range.

Reviewer 2 Report

The core step of the paper is a study on trajectories of source-space information flow extracted from resting-state magnetoencephalographic (MEG) data. The paper also includes the proposal of a new brain age index (BAI) that is employed in the study. The contribution of the paper is clear as it contains extensive experiments and discussions with significant clinical findings. Thus, results seem convincing. In general, the literal presentation of the paper is good, but there is still room for improvement in this regard. The methodology applied is adequate including time and frequency domains analyses. The correlation was used to estimate functional connectivity maps, but discussion on other approaches should be included. In summary, I consider the contents of the paper are potentially publishable, but the following minor issues should be addressed in a revised version of the paper.

- Literal presentation has room for improvement. For instance, (i) equation (6), the term dSTE should be defined before this equation, the same issue for equation (7) and dPE; (ii) please use the same font size and type in all equations; (iii) Some equations are outside the margins of the text. Equations must be aligned to the right. Therefore, an English proofreading of the paper is recommended.

- Page 3, lines 126-130, please provide a rationale of the selection of correlation for functional connectivity estimation. Besides, a theoretical or experimental comparison with other methods could be included, e.g., https://doi.org/10.1007/978-3-030-20518-8_38.

Author Response

The core step of the paper is a study on trajectories of source-space information flow extracted from resting-state magnetoencephalographic (MEG) data. The paper also includes the proposal of a new brain age index (BAI) that is employed in the study. The contribution of the paper is clear as it contains extensive experiments and discussions with significant clinical findings. Thus, the results seem convincing. In general, the literal presentation of the paper is good, but there is still room for improvement in this regard. The methodology applied is adequate including time and frequency domain analyses. The correlation was used to estimate functional connectivity maps, but discussion on other approaches should be included. In summary, I consider the contents of the paper to be potentially publishable, but the following minor issues should be addressed in a revised version of the paper.

Answer: We would like to thank you for your comments and the chance to review our study. We revised most parts of the study and the changes in the text are underlined in blue color.

- Literal presentation has room for improvement.

For instance, (i) equation (6), the term dSTE should be defined before this equation, the same issue for equation (7) and dPE;

Answer: You are right. We reorganize section 2.5 (old 2.4).

(ii) please use the same font size and type in all equations;

Answer: We rewrote the math equations in order to be consistent in terms of the font size.

(iii) Some equations are outside the margins of the text. Equations must be aligned to the right. Therefore, English proofreading of the paper is recommended.

Answer: We aligned math equations to the right place. The draft has been revised by two English native speakers.

- Page 3, lines 126-130, please provide a rationale for the selection of correlation for functional connectivity estimation. Besides, a theoretical or experimental comparison with other methods could be included, e.g., https://doi.org/10.1007/978-3-030-20518-8_38.

Answer: We added more text in the introduction part in order to familiarize the reader with the main core of this study, especially with the adapted estimators. We would like to avoid the comparison of the proposed methods with others for three reasons:  First of all, both estimators came from well-established mathematical theory and they have already been applied to many research studies. Secondly, we don’t aim to make the study exploratory in the sense of finding the ‘best’ estimator per amplitude and phase domain. Our findings showed significant age trends that were also repeatable. Third, we revised section 2.13 in order for a reader to understand the computational effort of the whole study.